# Biochemical Markers in Alzheimer’s Disease

**DOI:** 10.3390/ijms21061989

**Published:** 2020-03-14

**Authors:** Alessandro Rabbito, Maciej Dulewicz, Agnieszka Kulczyńska-Przybik, Barbara Mroczko

**Affiliations:** 1Department of Neurodegeneration Diagnostics, Medical University of Bialystok, 15-269 Białystok, Poland; maciejdulewicz@gmail.com (M.D.); kulczynska.agnieszka@gmail.com (A.K.-P.); mroczko@umb.edu.pl (B.M.); 2Department of Biochemical Diagnostics, Medical University of Bialystok, 15-269 Białystok, Poland

**Keywords:** Alzheimer’s disease, biomarkers, diagnosis

## Abstract

Alzheimer’s disease (AD) is one of the most frequent neurodegenerative diseases affecting more than 35 million people in the world, and its incidence is estimated to triple by 2050. Alzheimer’s disease is an age-related disease characterized by the progressive loss of memory and cognitive function, caused by the unstoppable neurodegeneration and brain atrophy. Current AD treatments only relieve the symptoms. The first molecular signs of the disease identified decades ago and were related to the tau neurofibrillary tangles and the β amyloid plaques. Despite the considerable progress in the diagnostic field, there is no certain knowledge of the specific biomarkers reflecting molecular mechanisms that trigger the symptoms of the disease. Therefore, there is an enormous need to find biomarkers useful for early diagnosis, before the first symptoms appear, and develop new therapeutic targets, which would guarantee improving patients’ quality of life. Researchers from all around the world are looking for biomarkers that can be identified in different biological fluids such as plasma, serum, and cerebrospinal fluid, specific for Alzheimer’s disease. In this review, we would like to resume some of the most interesting discovery in pathological mechanisms underlying Alzheimer’s disease and promising biomarkers.

## 1. Introduction

Alzheimer’s disease (AD) is a neurodegenerative disease and the most common cause of dementia, recognized by the World Health Organization as a global health public priority. Alzheimer’s disease was described for the first time in 1907 by the German psychiatrist and neuropathologist Alois Alzheimer [1]. Despite a long time having passed, many mechanisms of the disease are still not well understood. AD is one of the biggest challenges for modern neuroscience and medical diagnostics due to the vast complexity of progressing in the latent form of the neuropathological process. AD affects more than 35 million people in the world, and its incidence is estimated to triple by 2050 [2]; currently, the countries or regions with the most significant number of people affected are China and the developing western Pacific, western Europe, and the USA [3]. AD is a disease that occurs mainly in old age, in fact, many patients show symptoms after 65 years of age, but considering that at the moment we do not yet have the knowledge and/or tools necessary to make an early diagnosis from the onset of the disease, it is difficult to say precisely when pathological insults began. The most common form is Sporadic Alzheimer’s Disease (SAD), and about 5–15% of patients manifest Familial Alzheimer’s Disease (FAD). A distinction is also made regarding the period of Early Onset Alzheimer’s Disease (EOAD), that occur between 40 and 65 years of age [4,5] and a late form of the AD after the age of 65, also called Late Onset Alzheimer’s Disease (LOAD) [6]. Considering that, there is a lack of knowledge concerning what causes AD pathology and clear diagnostic criteria. Therefore, every day in the world, many researchers work on finding an answer to these two questions. New papers with encouraging results concerning the etiopathogenesis of the disease and new potential biomarkers, such as Neurogranin, NFL, CCL2, CXCL8, CXCL10, CXCL12, CCL5, CX3CL1, CXCL9, are continuously published. Moreover, in recent years, due to reviews and meta-analyses, we can have a clearer idea of promising biomarkers. In this review, we summarized the important pathological mechanisms leading to the development of AD and discussed candidate biomarkers, which could improve the diagnosis of AD.

## 2. Discussion

### 2.1. Mechanisms/Pathophysiology

The past 30 years of Alzheimer’s disease research have provided substantial evidence that the accumulation of abnormally folded proteins leads to neuronal death and, subsequently, the neurodegenerative process. The first discovered features of AD were amyloid plaques and neurofibrillary tangles (NFTs) (Figure 1), for this, the mechanisms most investigated as the cause of pathology were the amyloid cascade the NFTs cascade. These two are the most investigated hypotheses, but despite the years spent looking for the onset mechanism or treatment for the disease, based on these two most characteristic symptoms, there is still a lack of a cure. The current state of knowledge suggests that various hypotheses could explain the beginning of this disease. As already mentioned, the “classical hypothesis” are amyloid plaques and NFTs, among the most recent, there are the dopaminergic and the glymphatic system hypotheses. 

### 2.2. Amyloid Hypothesis

The main component of the senile plaques is β amyloid (Aβ) protein, which arises as a result of improper proteolytic processing of precursor protein (Amyloid Precursor Protein—APP), Aβ protein can exist in two different form: Aβ40, a protein, which does not cause pathological accumulations and Aβ42, the amyloidogenic protein, which is the main component of amyloid plaques [7]. The APP gene is located at chromosome 21q21 and encodes a type 1 transmembrane protein composed of three domains: The N-terminal long extracellular region, the short endothelial segment, and the short C-terminal segment found in the cytoplasm [8]. The APP protein is processed by the enzymatic complex of three secretases: α, β, γ. α-secretase converts APP to 83 amino acids soluble peptide from the C-terminal fragment of APP (CT83), a fragment with regulatory functions. β-secretase cleaves the N-terminal fragment with a length of 99 amino acids (CT99) [9], which remains bound to the cell membrane. γ-secretase through heterogeneous proteolysis metabolize CT83 and CT99 into the Aβ peptide in the form of Aβ40 and Aβ42. Aβ42 is the dangerous and neurotoxic form because it exhibits hydrophobic properties and is responsible for the formation of senile plaques [7,9]. Most microglial cells and astrocytes are responsible for the production of this form. Extracellular deposits of Aβ protein accumulate in areas that play an essential role in the control of memory and cognitive function. Several mutations in the APP gene were found to lead to EOAD, most of them were localized clusters in the Aβ encoding region of the APP gene. Furthermore, people with three chromosomes 21 copies, i.e., 21 trisomies and Down’s syndrome, developed AD neuropathology, but people with a partial 21 trisomy that did not include the APP gene did not develop AD neuropathology (4). Presenilin 1 (PSEN1), presenilin 2 (PSEN2), localizing in the endoplasmic reticulum are cofactors of the γ-secretase protein complex, are also responsible, albeit with different mechanisms and contributions, of the formation of neurotoxic forms of Aβ and the onset of the disease in people with (FAD) [10]. Presenilins are highly homologous proteins having 8–9 transmembrane tracts. In nerve cells, they act as membrane receptors and calcium channels, providing homeostasis, and their overexpression is most often observed in the hippocampus and Purkinje cells [11]. In the PSEN1 gene, 185 dominant mutations have been identified, which leads to approximately 80% of EAOD AD [12]. With a mutation in PSEN2, a fewer number of mutations were identified and caused approximately 5% of EAOD [4]. Besides mutations that lead to AD onset, there is also some genetic risk of factors for AD. The one most studied that is the strongest genetic risk factor for developing AD is APOE, particularly APOE4 isoform [13]. APOE is responsible for some different functions, including cholesterol transport, which can combine with causing the formation of deposits of the pathological form of Aβ, affect metabolism, and can be different ligand membrane receptors. Isoforms e2, e3, and e4, differ in cysteine and arginine residues at positions 112 and 158. One APOE4 allele imparts a threefold increase in risk, and two alleles impart a 12-fold increase in risk [14] to develop AD, and is also associated with EOAD [15,16]. Conversely, particularly the APOE2 isoform is associated with a lower risk of developing AD [17].

### 2.3. Neurofibrillary Tangles Hypothesis

The second most common aggregate in AD are the NFTs, found in various regions of the brain, composed mostly by paired helical filaments (PHF) of hyper-phosphorylated tau-protein [18,19]. The intracellular aggregation of hyper-phosphorylated tau-protein form causes the impairment of microtubule function, axonal transport, or the breakdown of the cytoskeleton of the nerve cell that leads to neurons degeneration. The activity of kinases and phosphatases, leading, respectively, to phosphorylation and dephosphorylation of tau-protein, affects the regulation of the balance between free tau-protein and microtubule-related protein. The imbalance between phosphorylation and protein dephosphorylation leads to impaired binding of this protein to microtubules and the formation of PHF and NFT. The structure of phosphorylated tau-protein may be influenced by Aβ, oxidative stress, neuroinflammation, and enzymes affecting kinases and phosphatases [20,21]. Three enzymes, which probably have the most important affect, are glycogen synthase kinase 3 (GSK3), cyclin-dependent Kinase 5 (CDK5), and microtubule affinity-regulating kinase (MARK) [22]. Several studies have shown that CDK5 is involved in tau-phosphorylation and NFTs progression [23,24]. Furthermore, it is very interesting that CDK5 may act as a regulator of GSK3, a kinase involved in AD onset too [25]. 

### 2.4. Glymphatic System Hypothesis

The human brain has four major fluids: Cerebrospinal fluid (CSF), interstitial fluid, intracellular fluid, and blood [26]. One of the recently discovered mechanisms associated with the removal of metabolites that prevent aggregation and deposition of amyloid beta is a glymphatic system (GS) [26]. The name glymphatic system comes from the analogous lymphatic system, with the difference that the work of the GS can be observed only in vivo. The glymphatic system plays an essential role in the clearance of the molecules from the brain parenchyma. The research was carried out mainly during slow-wave sleep, and provided the first direct evidence that the clearance of interstitial “waste products” increases during the resting state [27]. This investigation suggests that the effects of the GS depend, to a large extent, on the pressure and flow of CSF [28]. Markers circulating in CSF showed that the entry to GS was started in the parenchyma through a periarterial pathway surrounding the vascular smooth muscle cells bounded by the perivascular astrocytic endfeet [29]. The flow of CSF was associated with astrocytic water channels localized on endfeet and aquaporin 4 (AQP4) [29,30]. The proper functioning of GS is crucial for patients with AD and other neurodegenerative diseases dependent on abnormal metabolism. In a recent study, researchers suggested the disturbing expression of AQP4 was associated with aging and may be more susceptible to the aggregation of Aβ in the brain [31]. The most effective functioning of the glymphatic system takes place during sleep [29]. The disturbance of the circadian rhythm and sleep in AD patients may be an additional factor leading to improper functioning of the glymphatic system [26,32,33].

### 2.5. Dopaminergic Hypothesis

The dopamine system (DS) was associated with Parkinson’s disease, but its role is also being studied in AD. Patients manifest symptoms such as extrapyramidal signs and apathy that can be explained with a dopaminergic dysregulation [34,35]. A meta-analysis study showed that dopamine receptor 1 (D1) and dopamine receptor 2 (D2) concentrations were decreased in AD patients compared with the control [36]. Furthermore, it was shown that the treatment of AD patients with dopaminergic drugs such as Rotigotine, had beneficial effects [37].

Besides these results obtained in AD patients, recently, a study using an animal model of AD, investigated the effects of neuronal dopamine loss in memory and reward dysfunction [38]. In this study, using the Tg2576 mouse model of AD, overexpressing a mutated human APP, showed a loss of dopaminergic neurons in the ventral tegmental area before the plaques process occurred in the mice brains. This loss was observed in the ventral tegmental area, but not in the substantia nigra, which consequently reduced dopamine flow in the hippocampus and nucleus accumbens, brain areas involved in memory and reward [38].

### 2.6. Diagnosis of AD

Making an AD diagnosis is one of the significant challenges of modern medicine, not only in the early stage of the disease when the symptoms are not so evident but also in the most advanced stage and of dementia. This is because neurodegeneration begins a considerable amount of time before the onset of symptoms, but even when the symptoms of dementia were apparent, the certainty of diagnosis was achieved only by post-mortem autopsy. In recent years, the diagnostic criteria were updated by the National Institute of Aging (NIA) [39,40] and from the International Working Group (IWG-2) [41], based on biomarkers and imaging data. Therefore, it was necessary to use a multidisciplinary approach that included the use of imaging and clinical biochemistry methods accompanied by a neuropsychological analysis to assess the state of the patients. Biochemical and imaging diagnostic criteria are based on the main features of AD: Aβ plaques and tau NFTs. Radiolabeled molecule able to pass the blood-brain barrier and binds to Aβ plaques or tau NFTs were developed, thus allowing the evaluation of the aggregates in the human brain. The first molecule developed was the Pittsburgh compound B [42], able to bind the Aβ and that open the AD imaging era [43]. Before these molecules were synthesized, it was only possible to evaluate the atrophy of the brain through MRI [44]. Besides Pittsburgh compound B, other units were able to bind amyloid plaques that were developed and nowadays are in clinical use. Furthermore, a radiolabelled molecule was also developed that was able to bind tau aggregates. In addition to imaging techniques, the characteristic signs of the AD are also useful for the biochemical evaluation in biological fluids.

During biochemical tests, two isoforms of Aβ are evaluated: Aβ40, Aβ42, and two isoforms of tau protein: The phosphorylated tau isoform and the total tau protein concentration. Aβ42 in the cerebrospinal fluid (CSF) are widely accepted as biomarkers for AD. A lower level in CSF correlates with a higher presence of Aβ plaques in the brain [45]. These dependencies cannot explain how the reduction was also found in Creutzfeldt–Jakob disease [46] and cases of bacterial meningitis [47]. To improve the specificity of Aβ42 as a biomarker, a few years ago it was discovered that the ratio Aβ42/Aβ40 is significantly better than Aβ42 concentration to detect amyloid deposit in the brain mostly in the early stage of the disease [48]. Total tau protein in CSF, including phosphorylated and non-phosphorylated, is a common biomarker for neurodegeneration [49,50], but not specific for AD. Instead, CSF levels of phosphorylated tau seem to reflect the formation of neurofibrillary tangles in the brain [51,52].

To better interpret this information and be able to reduce the variability between different laboratories, a few years ago an algorithm was created that allows the interpretation of the CSF biomarkers of AD, ordering them on an ordinal scale from neurochemically normal people (score = 0) to probable AD (score = 4) [53]. This score, the Erlangen Score, can also be used to predict the evolution of mild cognitive impairment people (MCI) to dementia [54].

### 2.7. Promising Biomarkers

Given that AD causes massive neurodegeneration that begins years before symptoms are evident, in recent years, the search for potential biomarkers of brain damage that can diagnose AD at an early stage has started. Neurofilament light polypeptide (NFL) is a peptide encoded by the NEFL gene is an intrinsic cytoskeletal protein, that could be used as CSF and plasma biomarker of axonal damage. NFL is released after the neuronal death in the brain interstitial and then in the CSF and can reach the blood through arachnoid villi and perivascular drainage systems [55]. In a recent longitudinal study, it was found that the NFL level in serum of people that carried a mutation that led to the disease was elevated about 6.8 years before the first symptoms occurred. Furthermore, evaluating the NFL annual rate of change was possible to discriminate mutation carriers from non-carriers as early as 16 years before the estimated symptom onset. However, NFL cannot be considered a specific biomarker for AD. NFL is an essential component of the cytoskeleton of neuronal cells hence, an increased level was observed in other neurodegenerative disorders such as: Creutzfeldt-Jakob disease, amyotrophic lateral sclerosis, frontotemporal dementia, HIV-associated dementia, and others [56] [57].

Another protein that could be very useful in the early diagnosis of AD is neurogranin. This calmodulin-binding protein expressed primarily in the brain area most affected by AD as cortex and hippocampus, produced from excitatory neurons, and it is involved in protein kinase C [58,59]. In a study conducted a few years ago, the levels of neurogranin in the CSF were compared in people with AD, frontotemporal dementia, Lewy body dementia, Parkinson’s disease, and multiple system atrophy. In this study, neurogranin concentration in CSF was higher in AD comparing to the control group, and also showed that a higher neurogranin concentration in CSF was specific for AD, as a higher concentration was not found compared to the control in the other neurodegenerative diseases [60].

Potential biomarkers for AD are not just biomarkers of neurodegeneration, indeed several years ago, this was demonstrated using animal models and clinical studies that AD played a central role in the inflammatory process. Systemic inflammation in AD might exacerbate the neurological deficit, in fact, inflammatory molecules can contribute to blood-brain barrier permeability, therefore, in the last years, a significant number of studies that investigate the alteration in peripheral inflammatory molecule concentration in AD have come out. A recent study examined the circulating levels of Il-1 family cytokines and their receptors, measuring the concentration of 12 proteins of this family in serum [61]. Results of this research found that inflammatory Il-1 family cytokines were mostly increased in Alzheimer’s disease patients compared with the control, particularly the increase of the anti-inflammatory cytokines member of this family correlate with the small increase of the pro-inflammatory cytokines as Il-1α and Il-1β. As the authors suggest, it was possible to hypothesize that there could be a physiological mechanism of response to the inflammatory environment that occurs with the disease, evidently though with poor results, although an increase in the concentration of the soluble receptors sIL-1R1 and sIl-1R3, but not an increase of sIL-1R2 [60]. Il-1β as a pro-inflammatory cytokine has already been studied as a possible biomarker in Alzheimer’s disease by different research groups, and although there were differences in the concentrations found, the common trend was that there was an increase in the concentration of this pro-inflammatory cytokine to confirm that in concomitance with protein aggregations there was inflammation in the brain that was also found in peripheral fluids [62,63,64]. In the database there is a large number of articles investigating the change in concentration of chemokines in plasma or serum, some results, they talked to each other, but we can say thanks also to the meta-analyses carried out on the subject that the dysregulation of these proteins is in progress in patients with AD. Many chemokines concentration was assessed in biological fluids of Alzheimer’s disease patients, comparing the resulting concentration with a control group or MCI group. To evaluate whether these molecules involved in inflammation act in the central nervous system or at the peripheral level, the concentrations in peripheral fluids such as blood, serum, or plasma are measured, as well as in CSF (Table 1). Zhang et al. investigated the concentration of CCL2 protein, also known as monocyte chemoattractant protein 1 (MCP1), in the serum and CSF, and in both fluids, CCL2 was increased in AD patients compared with the control group [65]. Other authors revealed that similar to CCL2, the concentration of CXCL8, previously called IL-8, was also increased both in the serum and CSF of patients with AD [66,67]. Interestingly, the levels of CXCL10 (Interferon gamma-induced protein 10 (IP-10) or small-inducible cytokine B10) [66] was increased only in the CSF of AD patients, whereas CXCL9 (monokine induced by gamma interferon (MIG)) concentration was higher in the whole blood of AD patients [68]. A different direction of concentration changes was observed for CXCL12, also known as SDF-1, CCL5 [69], and CX3CL1. Decreased levels were observed of CXCL12 in the serum and CSF [70], whereas CCL5 [69] and CX3CL1 [71] only in the serum. Fluctuations of the concentrations of the above-mentioned proteins and these differences between various chemokines possibly are related to the different roles they play. Furthermore, based on current knowledge, we are not able to clearly confirm that the alterations of the levels of above-discussed chemokines in peripheral blood are related to CNS neurodegeneration. It is also possible that these changes in the concentration are caused by systemic disease or aging.

## 3. Conclusions

Alzheimer’s disease affects a large percentage of older people and constitutes the fifth leading cause of death [72]. Although the first case of this disease was described more than a century ago, at the present moment, the exact pathological mechanism of AD is still unknown. Therefore, it is a major challenge both for biomedical and clinical research. Although the causes of the disease are not known, the most important molecular signs have been described as amyloid plaques and neurofibrillary tangles. The accumulation of these proteins triggers cell death, resulting in incessant and irreversible neurodegeneration. Currently, based on experimental evidence, various hypotheses have been formulated that can explain the onset of the disease, including hypothesis of the amyloid, the neurofibrillary tangles, the dopaminergic, and the glymphatic system. However, these hypotheses at the moment would seem to be all plausible, and we should not underestimate the role played by pro-inflammatory molecules that create an inflammatory environment in the brain that could be a direct consequence of the disease, or that could contribute to its onset. The research efforts are not only aimed at understanding the causes of AD but also to improve diagnostics tests for early recognition of this disease. Although the neurodegenerative process begins a long time before the first symptoms appear, the current treatment can only relieve AD symptoms. The AD diagnosis is carried out with a multidisciplinary approach. The following aspects are evaluated: The neuropsychological condition of the subjects, the pathological changes in the brain by using imaging tests, as well as the presence of specific the pathology proteins with biochemical tests. The measuring levels of Aβ40 and Aβ42, phospho tau and total tau had fundamental importance in the diagnosis of AD, but probably more important was the implementation of the Erlangen score, and the Aβ40/Aβ42 ratio in order to evaluate the stage of the disease. Due to insufficient specificity of classical biomarkers, researches look for some novel biomarkers useful for early diagnosis, including, e.g., some inflammatory molecules and synaptic proteins. Despite the great progress in the biochemical markers research, there are still no certainties about the diagnosis and particularly about the cause of this disease. Giving that AD is a heterogeneous and multifactorial disorder, it seems that diagnosis should be based on the analysis of various proteins, reflecting different pathological mechanisms, combined with imaging and neuropsychological examination. Therefore, it is necessary to create a panel of specific proteins, which concentrations will be changed with the stage of the disease. Because of the above-mentioned difficulties, it will be crucial to devise computational models that can predict the evolution of the pathology using machine learning (ML). Research teams need to collaborate to create accurate models of the disease. Investigations of concentration changes of biomarkers would provide necessary data to develop a model of the disease by using ML and artificial intelligence (AI), which may provide a more accurate diagnosis and information on the progression of the disease. In the future, AI and ML models based on biochemical, neuroimaging, and life trackers data will be increasingly important for prognosis of the AD course.

Data concerning the concentrations of novel biomarkers are inconsistent. It can be related to many factors, among others, different detection methods, lab procedures, problems with standardization of tests and establishing cut-off values. All these difficulties and hurdles may be overcome using artificial intelligence. Models of AI and ML based on a large amount of data will allow the identification of the most specific and sensitive biomarkers for the AD.

## Figures and Tables

**Figure 1 ijms-21-01989-f001:**
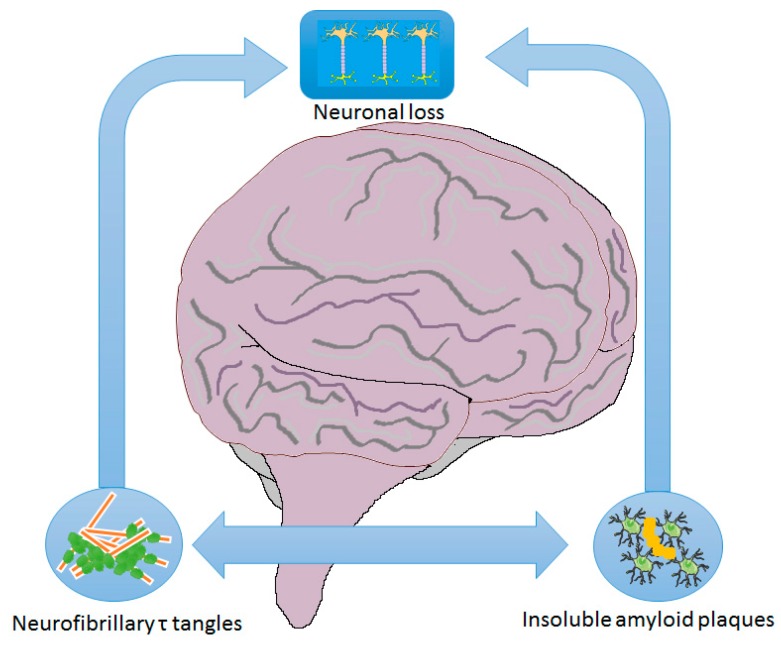
Major Alzheimer’s disease molecular signs.

**Table 1 ijms-21-01989-t001:** Promising Alzheimer’s disease biomarkers variations. EA = AD early stage, NA = Not available information about the stage of the disease or MMSE. * [56].

Sample	Protein	AD vs. Control	MMSE	Ref.
CSF	Neurogranin	↑ (NA)	AD (median) 21 (17–25) HC (median) 30 (29–30)	[60]
CSF	CCL2	↑ (EA)	AD (mean) (24.5 ± 2.1)	[65]
CSF	CXCL8	↑ (NA)	NA	[66]
CSF	CXCL10	↑ (NA)	NA	[66]
CSF	CXCL12	↑ (EA)	AD (mean) (18.9 ± 4.1)	[70]
Serum	CXCL12	↑ (NA)	AD (mean) (23.6 ± 1.6) CTRL (mean) (28.4 ± 1.6)	[70]
Serum	NFL	↑ (NA)	*AD (mean) (23.2 ± 2.1) *CTRL (mean) (29.1 ± 1.0)	[56,57]
Serum	CCL5	↓ (NA)	NA	[69]
Serum	CX3CL1	↓ (NA)	AD (mean) (15.3 ± 3.6) CTRL (mean) (27.3 ± 1.0)	[71]
Whole Blood	CXCL9	↑ (NA)	NA	[68]

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
