# Peer review of "Biochemical Markers in Alzheimer’s Disease"

_ijms, 2020, doi:10.3390/ijms21061989_

Round 1
Reviewer 1 Report
In this manuscript, authors mainly organized many biomarkers which could be possibly used to detect Alzheimer's disease (AD) from previous papers. To detect AD in early stage, the specific biomakers of AD are very important and they will largely affect the targeting confidence. Although no suitable strategies to detect AD in early stage until now, the development of AD diagnosis is urgent This manuscript can provide useful information for developing AD-detected platforms. Therfore, I recommand this manuscript can be published in IJMS after minor revision.
My questions for this manuscript are as below.
1. Please insert "Table 1" into content.
2. Please identify the CCL2, CXCL8 and the other similar proteins. For example, CCL2 is monocyte chemotactic protein 1.
3. I suggest authors add the stages of higher or lower expression (compared to control) for each protein in Table. This will make readers clearly understand what proteins can be detected in different stages of AD.
Author Response
Dear Reviewer,
Please consider this revised manuscript entitle “Biochemical markers in Alzheimer’s disease” for the publication in the International Journal of Molecular Science. We have made the changes requested by you, the answers are presented below.
- Please insert "Table 1" into content.
For this point, we add “Table 1” in the text.
- Please identify the CCL2, CXCL8 and the other similar proteins. For example, CCL2 is monocyte chemotactic protein 1.
For the point #2, we add in the text the alternative name of the proteins, although nowadays it should be more correct to use the nomenclature based on the amino acid sequence of chemokines.
- I suggest authors add the stages of higher or lower expression (compared to control) for each protein in Table. This will make readers clearly understand what proteins can be detected in different stages of AD.
Fort this last point we add in the table the AD stage where applicable. Unfortunately, not all papers highlight the progress of the disease, considering that neurodegeneration begins years before symptoms appear, it is very difficult to establish at what stage of the disease the patient is at the time of biochemical analyzes. Additionally, we add also the information about the MMSE in the table.
We hope that we have understood the intentions and suggestions of the reviewers and have corrected manuscript properly. Furthermore, the abstract and conclusion have been modified with the aim of better summarizing the key-points of the manuscript.
Best regards,
Alessandro Rabbito
Reviewer 2 Report
In this review Biochemical markers in Alzheimer’s disease, the authors have clearly outlined the current biomarkers in predicting the disease.
The review requires an elaborate introduction stating what led to writing a review.
The Review also requires to state what are the current pitfalls in coming out with biomarkers and the difficulties and hurdles that need to overcome.
The review also needs to show what led to writing and how it has added value in terms of the missing information.
The future directions in research that is necessary to come up with predictable biomarkers and will they help in early detection.
Author Response
Dear Reviewer,
Please consider this revised manuscript entitle “Biochemical markers in Alzheimer’s disease” for the publication in the International Journal of Molecular Science. We have made the changes requested by you, the answers are presented below.
Answer the questions 1 and 3
The review requires an elaborate introduction stating what led to writing a review; the review also needs to show what led to writing and how it has added value in terms of the missing information.
Regarding this 2 points we add the following text in the manuscript:
Considering that, there is a lack of knowledge concerning what causes of the AD pathology and clear diagnostic criteria. Therefore, every day in the world, many researchers work on finding an answer to these two questions. New papers with encouraging results concerning the etiopathogenesis of the disease and new potential biomarkers, such as Neurogranin, NFL, CCL2, CXCL8, CXCL10, CXCL12, CCL5, CX3CL1, CXCL9, are continuously published. Moreover, in recent years, due to reviews and meta-analyses, we can have a clearer idea of the promising biomarkers. In this review, we summarize the important pathological mechanisms leading to development of AD and discussed candidate biomarkers, which could improve the diagnosis of AD.
Answer the questions 2 and 4
The Review also requires to state what are the current pitfalls in coming out with biomarkers and the difficulties and hurdles that need to overcome; The future directions in research that is necessary to come up with predictable biomarkers and will they help in early detection.
Regarding this 2 points I add the following text in the manuscript:
Due to insufficient specificity of classical biomarkers, researches look for some novel biomarkers useful for early diagnosis, including e.g. some inflammatory molecules and synaptic proteins. Despite of the great progress in the biochemical markers research there are still no certainties about the diagnosis and particularly about the cause of this disease. Giving the AD is heterogeneous and multifactorial disorder it seems that diagnosis should based on the analysis of various proteins, reflecting different pathological mechanisms, combined with imaging and neuropsychological examination. Therefore, it is necessary to create a panel of specific proteins, which concentrations will be change with the stage of the disease. Because of the above mentioned difficulties, it will be crucial to devise computational models that can predict the evolution of the pathology using machine learning (ML). Researcher teams need to collaborate, to create accurate models of the disease. Investigations of concentration changes of biomarkers would provide necessary data to develop model of the disease by using ML and artificial intelligence (AI), which may provide a more accurate diagnosis and information of progression of disease. In the future, AI and ML models based on biochemical, neuroimaging and life trackers data will be increasingly important for prognosis the AD course.
Data concerning the concentrations of novel biomarkers are inconsistent. It can be related to many factors, among others, different detection methods, lab procedures, problems with standardization of tests and establishing cut-off values. All these difficulties and hurdles maybe overcome using of artificial intelligence. Models of AI and ML based on a large amount of data will allow to identify most specific and sensitive biomarkers for the AD.
We hope that we have understood the intentions and suggestions of the reviewers and have corrected manuscript properly. Furthermore, the abstract and conclusion have been modified with the aim of better summarizing the key-points of the manuscript.
Best regards,
Alessandro Rabbito